# Evaluating Outcome Measure Data for an Intensive Interdisciplinary Home-Based Pediatric Feeding Disorders Program

**DOI:** 10.3390/nu14214602

**Published:** 2022-11-01

**Authors:** Meeta R. Patel, Vipul Y. Patel, Ashley S. Andersen, Aida Miles

**Affiliations:** 1Clinic 4 Kidz, P.O. Box 1711, Sausalito, CA 94966, USA; 2Department of Pediatrics, Division of Gastroenterology, School of Medicine, Stanford University, Palo Alto, CA 94304-5660, USA; 3Department of Pediatrics, School of Medicine, University of Alabama, 1600 7th Avenue South, Birmingham, AL 35233-1771, USA

**Keywords:** feeding tube dependence, failure to thrive, food selectivity

## Abstract

Background: The purpose of this study was to evaluate the effectiveness of an interdisciplinary home-based feeding program, which is a unique service delivery model. Methods: Data were provided on oral intake, tube feeding elimination, and weight for patients who were dependent on tube feedings (*n* = 78). Weight data were collected for patients who showed failure to thrive (*n* = 49). Number of foods consumed and percentage of solids were collected for patients who were liquid-dependent (*n* = 23), and number of foods consumed were collected for patients who were food-selective (*n* = 61). Results: Data were analyzed using paired sample *t*-test with 95% confidence interval. For patients dependent on tube feedings, 81% achieved tube feeding elimination. Tube elimination was achieved after 8 months of treatment on average. All failure-to-thrive patients showed weight gain from baseline to discharge. For liquid-dependent patients, there was an increase in foods consumed from 2 foods at admission to 32 foods at discharge. For food selective patients, there was an increase from 4 foods at admission to 35 foods at discharge. For all dependent variables, results showed statistical significance and a large-sized effect. Conclusions: These data show that an intensive interdisciplinary home-based program can be successful in treating complex feeding problems in children.

## 1. Introduction

A pediatric feeding disorder has been defined as impaired oral intake that is not age-appropriate, and is associated with medical, nutritional, feeding skill, psychological dysfunction, or a combination of these factors [1]. Feeding problems can be classified into four distinct categories (1) eating nothing to minimal amounts by mouth (PO), (2) poor oral intake, (3) liquid dependence, and (4) food selectivity.

Children who eat minimally or nothing PO may require enteral tube feedings to supplement or provide full nutritional support [2,3,4]. Children who have poor oral intake may not be dependent on tube feedings; however, they are at risk for tube placement, failure-to-thrive (FTT), or other growth or nutritional deficiencies [5,6]. Children who are liquid-dependent (LD) are those who eat no solid foods or limited volumes of solids and most of their nutrition and calories come from a liquid substance [7]. Food-selective (FS) children often have a limited repertoire of foods they consume or are dependent on lower textured foods, such as jarred baby food or pureed foods, at inappropriate ages [8,9,10,11,12]. 

A pediatric feeding disorder may develop from medical complications, anatomical abnormalities, negative experiences with eating, or a combination of these factors [13,14]. It is likely that, even when medical and anatomical issues have been resolved by medication or surgeries, food refusal persists. Food refusal or avoidance may occur because, over time, the child learns that inappropriate mealtime behavior results in the termination of meals. In the long term, if the child does not eat, skill deficits and sensory sensitivities become evident [14].

Treatment for feeding problems is initially provided by individual professionals with expertise in different areas, such as medicine, speech or occupational therapy, dietetics, and applied behavior analysis (ABA) or psychology using an outpatient model and is typically provided during weekly appointments. However, when the feeding problem is more severe, the treatment of choice is the use of an inter-multidisciplinary team, where expertise in all these areas is brought together to develop a comprehensive treatment plan using an intensive all-day treatment model [15].

Numerous studies have demonstrated the effectiveness of treatments conducted within inpatient or day-treatment multidisciplinary pediatric feeding disorders programs using behavioral treatment models [16,17,18], which can range from 3- to 8-week programs. Although these programs have been shown to be effective, the long-term success of these children is unknown with regard to parameters such as eating like age-typical peers or eating in a variety of settings. In addition, obtaining insurance approval for these services has been challenging due to their cost. 

The average cost of treatment in these programs can be about USD 130,380.00, including evaluation, inpatient treatment, and outpatient follow-up care [18]. However, the cost is nowhere near the costs of maintaining a child on gastrostomy (G-) tube feedings. The mean yearly adjusted cost for maintaining children on G-tubes is about USD 46,875.55, with a five-year total cost of USD 234,377.75 [17]. These costs are only for uncomplicated care and do not include additional medical appointments or hospitalizations related to the tube. Nevertheless, there is a cost saving in the long term when treatment is provided in an inpatient or day-treatment multidisciplinary pediatric-feeding-disorder program. 

Treatment can also be delivered in the home; however, there is a dearth of data utilizing a home-based model to treat feeding problems. Some studies have shown the effectiveness of a home-based tube weaning program where either hunger induction or behavioral treatments were evaluated [19,20]. Although the data show a successful transition to PO feeds, weight loss was observed when hunger induction was the focus of treatment and long-term weight-gain data were not presented [19]. Additionally, during follow-up, tube feedings were only eliminated for 67% of patients [20]. Moreover, previous studies have not presented data on age-typical eating patterns at discharge. 

The aforementioned studies focused only on children dependent on enteral feedings; therefore, the impact of a home-based model on other feeding problems is unknown. One study presented short-term behavioral treatment for children with food selectivity and showed increases in the number of foods consumed, and parents were trained to implement the treatment protocol successfully [12]. These data are promising; however, an inter- or multidisciplinary team was not utilized. In addition, previous studies examining a home-based model have not presented data on long-term success and cost–benefit data. To our knowledge, there have been no outcome data published evaluating an intensive home-based interdisciplinary feeding program using a broader population. The purpose of this study was to evaluate the outcomes of patients who were admitted to our Intensive Interdisciplinary Home-Based Pediatric Feeding Disorders Program where all disciplines worked together to come up with a comprehensive treatment plan to eventually achieve age-typical eating patterns. 

## 2. Materials and Methods

### 2.1. Participants

A retrospective cohort study was conducted through a chart review of children sequentially admitted to a home-based pediatric feeding disorders program. The program was created by the first author in 2003 to serve families in a geographic area lacking intensive feeding programs. Upon admission, families signed a consent form to allow for outcomes data to be collected, analyzed and presented; IRB (Institutional Review Board) review was not solicited.

Only patients who were in this treatment for at least 12 months or met all goals prior to the 12 months were included in this study. Of the 215 patients admitted, only 24 were prematurely discharged and were not included (see Figure 1). This study included 191 participants with various medical diagnoses (see Table 1). The majority of patients had private insurance through their employer. As for race, 56% of patients were white and 44% were non-white. As for gender, 42% were female and 58% were male. Age ranged from 10 to 144 months (M = 44; SD = 25.8). All patients lived at home with at least one biological or adoptive parent.

Patients were categorized by feeding condition: (a) tube-feeding-dependent (*n* = 78, TFD), (b) FTT (*n* = 49), (c) LD (*n* = 23), (d) FS (*n* = 61). Some patients fell into more than one condition based on presenting problems. The number of participants (*n*) varied across time intervals due to unavailable data (see Tables 2–5). 

### 2.2. Setting

Treatment was provided in the patient’s home, school, and other environments (e.g., restaurants, park) with initially therapist conducted meals and after caregiver training, family-conducted meals.

### 2.3. Treatment

The Intensive Interdisciplinary Pediatric Feeding Disorders Program included a dietitian, speech and language pathologist (SLP), a Board-Certified Behavior Analyst-Doctoral (BCBA-D), and a case manager who was typically a BCBA with a master’s degree in Applied Behavior Analysis or Psychology. The referral and treatment process is outlined in Figure 2.

Prior to intensive treatment, a 2–3 h evaluation was conducted in the home by the BCBA-D to evaluate behavioral and oral motor dysfunction. A medical and feeding history was obtained during the evaluation, and a caregiver conducted a typical meal while the clinician observed. After the observation, specific recommendations were provided to the family. The rest of the team was consulted after the evaluation and further recommendations were provided to the family, if needed. Thereafter, there was a weekly check-in with the family via phone or email until treatment was initiated.

Thereafter, the case manager coordinated a treatment plan with the feeding team. The feeding team provided in-person training to the case manager with regards to dietary modification, oral motor exercises, behavioral treatments, and safety measures. The same case manager provided treatment to the child from admission to discharge and the patient’s existing medical team provided consultation on an as-needed basis. 

The initial treatment visit was 3–5 consecutive days. The case manager was in the patient’s home from breakfast to dinner, typically 10–12 h per day with a lunch break in between these hours for the case manager. A meal schedule was developed by the case manager and the family based on what would work in their daily routine, even in the absence of our physical visits. Typically, patients had meals 4×/day (breakfast, lunch, snack, and dinner), which lasted up to 30 min in a set location in the home. The duration of meals was individualized to each patient. Initially, multiple short (5–10 min) sessions may be conducted by the case manager so the child could get used to the structured schedule. 

Prior to conducting any meal sessions, the case manager built a rapport with the child by playing with them. This type of rapport-building occurred in between meals as well to ensure that the child had a positive relationship with the case manager and that trust could be built even outside of mealtimes. Trust is crucial when it comes to teaching children to eat, especially if eating has been paired with negativity in the past. This rapport building also allowed for the case manager to get to know the child and determine what type of positive reinforcement (e.g., toys, music, videos, preferred food, etc.) could be used in treatment. After rapport-building occurred, specific goals for the visit were set. In fact, goals for the entire admission were set sequentially based on baseline performance and what steps were needed to eventually get to age- or developmental-typical eating patterns. Goals were individualized to the patient and based on what the patient was doing at the time of admission. Initially the goals for most patients who were admitted were to decrease refusal behaviors and increase acceptance. Empirically validated behavioral treatments were implemented with consultation with the BCBA-D [21,22,23,24,25,26,27,28]. For example, if the child was not accepting any food at the start of treatment, the goal would be to conduct meal sessions based on acceptance of an empty spoon or whatever the child was willing to accept and slowly adding food to the spoon. Treatment always started with antecedent strategies, developing a starting point for treatment based on baseline levels of responding [29]. Antecedent strategies were employed to keep refusal behaviors lower; however, antecedent strategies alone were often not effective. In addition, to antecedent-based interventions, consequence-based treatments were also implemented, such as positive reinforcement and escape extinction [21,22]. These treatments were implemented to decrease food refusal and increase acceptance. In addition, similar treatments were used to build oral motor skills as well. For example, positive reinforcement and modeling were used to teach a child to first chew on a chew stick, then eventually on food [30]. The oral motor progression was developed with the assistance of the SLP. The SLP also assisted in determining safety measures for swallowing and seating arrangement. In addition, treatment also involved dietary modifications in consultation with the dietitian.

Initially, the case manager conducted treatment sessions and data were collected on a variety of measures (i.e., food acceptance, total consumption, food refusal, swallowing, chewing, tongue lateralization). A structured protocol was used during all meal sessions based on the specific behavioral treatment or oral motor progression. In addition, specific food type and volume was based on the meal plan. Once an effective treatment was developed, caregivers were trained to follow the treatment plan. Caregivers were trained using a modified behavioral skills training model [31,32], where they were given a written treatment plan, modeling, rehearsal, and immediate feedback while feeding their child. At the end of the 3–5 days, a meal was conducted through telehealth to ensure the patient ate for caregivers in the absence of the case manager. Caregivers were required to continue to conduct meals 4×/day in between our physical visits according to the final treatment protocol for the visit. They were also trained to collect data to monitor progress in between physical visits. After the initial treatment, the case manager contacted the family on a weekly basis to make adjustments to the treatment plan if necessary. 

Follow-up visits were identical to the initial treatment visit and were initially conducted 3 days per month. The case manager was also in contact with the family weekly or as needed in between physical follow-up visits. During every follow-up, we worked on 2–3 goals per month and caregivers continued to be trained every month on the final treatment protocol. In addition, school training was provided during follow-up to ensure that the patients ate at school and that there was continuity from home to school. School treatment plans were similar to home; however, some modifications may have been made based on location and staffing. As the patient met their goals, visits were reduced to 2 days per month, and then every 1–3 months until discharge. During follow-up visits, we continued to work on goals to eventually achieve age- or developmentally typical eating. The family was trained to conduct meals in all locations, and components of the treatment plan faded over time and until the child ate without a structured treatment plan. 

### 2.4. Dependent Measures and Data Analysis

Data were collected on the percentage of each patient’s calorie goal that was met for TFD patients. The dietitian determined the calorie goal for each patient based on age, weight, height, and history, which served as the denominator. Data were collected on the number of calories consumed at different intervals in treatment which served as the numerator for the calculations. *Baseline* refers to average calorie goal consumed at admission based on a 3-day food diary obtained from the caregiver. *After Initial Treatment* refers to the calorie goal consumed during the last day of initial treatment. Subsequent intervals refer to calorie goal consumed on average during the visit at that interval based on data completed by the case manager. *Discharge* refers to the calorie goal consumed on the last day of the treatment program. Discharge time varied per patient because it was based on when goals were met.

Data were collected on the percentage of patients who were 100% PO at discharge. The data for mean months on tube feedings prior to treatment and mean months before tube elimination were also collected. Data were collected on the percentage of weight gain at discharge for TFD patients who were weaned from tube feedings during treatment. Weight data were collected at home by the caregiver, under the same conditions at each interval. Weight at one month was compared to the weight at baseline to determine the percentage increase. Subsequent weights were compared to the previous time interval; however, discharge weight was compared to baseline weight. The mean percentage increase in weight gain across patients was calculated. 

We collected data on percentage increase in weight gain across different intervals in treatment for FTT patients, similar to TFD patients. Weight data were collected at different treatment intervals; however, for some patients, only baseline and discharge weights were available due to a lack of working equipment in the home. 

Data on the number of foods consumed at each interval were collected for LD and FS patients. The food was considered “consumed” if the child was eating at least 1 oz of that food. We also collected data on the percentage of solids consumed at baseline and discharge for LD patients. The percentage of solids consumed was calculated by dividing the calories consumed from solids divided by the patient’s total caloric intake. The mean number of foods and mean percentage of solids consumed were calculated across patients. 

Data were also collected on patients receiving prior therapies. We calculated the percentage of patients who received another feeding therapy prior to our treatment program and the average length of treatment in those therapies. Data on the number of months to discharge from our program and the average number of visits before discharge were also collected. 

Data were collected on the percentage of patients who achieved “typical” eating patterns at discharge. To be considered a “typical” eater, the patient must have met the following criteria: (1) self-fed all meals if developmentally appropriate, (2) ate in two other locations besides the home, (3) ate at least two foods in each food group at age appropriate portions, (4) refusal behaviors occurred for less than 10% of the meal, and (5) ate regular textured food or the texture that was appropriate for age or development. The feeding team evaluated whether age or developmental feeding norms were met. 

Analyses were conducted using IBM (Armonk, NY, USA) SPSS Statistics software, Version 27. A series of paired sample *t*-tests were used to assess whether there was a significant change in each outcome from baseline to the final assessment timepoint (in most cases, 12 months). A *p-* value less than 0.05 was considered significant. Cohen’s *d* value was used to determine the size of the effect. 

### 2.5. Treatment Satisfaction

An anonymous 19-question Likert scale survey was given to families during the last day of treatment to assess treatment satisfaction. Caregivers were instructed to fill out the survey together to ensure that both caregivers agreed with the answers. Only one survey was given to each family to fill out. Caregivers rated questions from 1–5, 1 being quite dissatisfied and 5 being extremely satisfied. An average score was calculated across patients. 

## 3. Results

Data were analyzed using paired-sample *t*-test (see Table 2, Table 3, Table 4 and Table 5) with 95% confidence interval. For TFD patients, the mean percentage of calorie goal coming from PO feedings at baseline was 20% (see Table 2). After 12 months of treatment, the mean percentage of calorie goal coming from PO feedings was 109%, exceeding the recommended goal. The number of calories consumed PO was significantly higher at 12 months of treatment (M *=* 1200.73, SD = 456.555) compared to baseline (M = 203.43, SD = 313.613; *t* (52) = −18.001, *p* < 0.001). Cohen’s *d* (2.473) indicated that this was a large-size effect. (Cohen’s *d* range; <0.2 = small-effect; 0.2 to 0.5 = medium-effect; >0.8 = large-effect). A paired *t*-test was conducted comparing baseline to after initial treatment and to 3 months; the results showed statistical significance and a large-sized effect.

**Table 2 nutrients-14-04602-t002:** Tube Feeding Dependent—Percentage of kcal Goal Met PO.

Time	*n* ^a^	Mean % (SD) ^b^	% Typical Eating ^c^
Baseline	78	19.9 (0.247)	
After Initial Tx	78	52.8 (0.379)	
3 Months	73	80.7 (0.381)	
6 Months	72	91.9 (0.395)	
9 Months	57	101.5 (0.371)	
12 Months	53	108.8 (0.354)	
Discharge (Typical Eating)	72	-	72.2

Data were analyzed using a paired-sample *t*-test with 95% confidence interval. The first column represents the time interval. Tx= treatment. ^a^ Number of patients during that time interval. ^b^ Mean percent of calories goal met PO by patients at that time interval; Standard Deviation (SD) in parenthesis. ^c^ Percent of patients who were tube-feeding-dependent upon admission that achieved typical eating at discharge. Patients may be counted in more than one category by feeding condition if they were admitted with multiple feeding problems. Calorie data were no longer collected at discharge because progress was monitored by volume consumed. PO= by mouth.

**Table 3 nutrients-14-04602-t003:** Failure to Thrive—Percentage Increase in Weight.

Time	*n* ^a^	Weight (SD) ^b^	Mean % (SD) ^c^	% Typical Eating ^d^
Baseline 1 Month	49 44	25.3 (6.850) 25.8 (5.966)	N/A 4.3 (0.038)	
3 Months	38	26.4 (5.850)	4.6 (0.039)	
6 Months	33	28.1 (6.324)	6.4 (0.045)	
9 Months	30	29.1 (6.813)	4.5 (0.050)	
12 Months	24	29.6 (6.250)	5.0 (0.042)	
Discharge	49	33.3 (7.890)	35.2 (0.302)	
Discharge Typical Eating	44		-	84.1

Data were analyzed using a paired-sample *t*-test with 95% confidence interval. The first column represents the time interval. Some patients only had baseline and discharge data. ^a^ Number of patients during that time interval. ^b^ Mean weight at each interval in pounds; Standard Deviation (SD) in parenthesis. ^c^ Mean percent increase in weight gain for patients from the previous interval. ^d^ Percent of patients who had a failure to thrive diagnosis upon admission that achieved typical eating at discharge. Patients may be counted in more than one category by feeding condition if they were admitted with multiple feeding problems.

**Table 4 nutrients-14-04602-t004:** Liquid-dependent—Number of Foods Consumed.

Time	*n* ^a^	Mean (SD) ^b^	% Typical Eating ^c^
Baseline	23	2 (2)	
After Initial Tx	23	14 (7)	
1 Month	20	22 (10)	
2 Months	17	27 (12)	
3 Months	17	28 (13)	
4 Months	13	28 (15)	
5 Months	11	28 (13)	
6 Months	7	28 (11)	
Discharge	22	32 (17)	
Discharge Typical Eating	20	-	85.0

Data were analyzed using a paired-sample *t*-test with 95% confidence interval. The first column represents the time interval. Some patients only had baseline and discharge data. ^a^ Number of patients during that time interval. ^b^ Mean number of foods consumed by patients at that time interval; Standard Deviation (SD) in parenthesis rounded to the nearest whole number. ^c^ Percent of patients who were liquid-dependent upon admission that achieved typical eating at discharge. Patients may be counted in more than one category by feeding condition if they were admitted with multiple feeding problems. Number of foods consumed were no longer collected at each interval after 6 months. Thereafter, only the discharge numbers of foods consumed were collected.

**Table 5 nutrients-14-04602-t005:** Food Selective—Number of Foods Consumed.

Time	*n* ^a^	Mean (SD) ^b^	% Typical Eating ^c^
Baseline	61	4 (4)	
After Initial Tx	61	14 (7)	
1 Month	47	19 (7)	
2 Months	41	21 (8)	
3 Months	37	24 (9)	
4 Months	31	24 (9)	
5 Months	34	27 (11)	
6 Months	32	29 (12)	
Discharge	44	35 (19)	
Discharge Typical Eating	60	-	85.0

Data were analyzed using a paired-sample *t*-test with 95% confidence interval. The first column represents the time interval. ^a^ Number of patients during that time interval. ^b^ Mean number of foods consumed by patients at that time interval; Standard Deviation (SD) in parenthesis rounded to the nearest whole number. ^c^ Percent of patients who were food-selective upon admission that achieved typical eating at discharge. Patients may be counted in more than one category by feeding condition if they were admitted with multiple feeding problems. Number of foods consumed were no longer collected at each interval after 6 months. Thereafter, only discharge numbers of foods consumed were collected.

The percentage of TFD patients who met 100% PO, including calories, fluids, and medication, was 81%. Tube eliminations varied by medical diagnoses (see Table 1). Some failures in this category were attributed to still needing the tube for fluids or medications upon discharge but all calories were being consumed PO. The mean months patients had a feeding tube prior to treatment was 27 months. There was a 29% increase in weight between baseline and discharge for patients who were previously dependent on tube feedings. This increase reflects weight gain in the absence of tube feedings. 

For FTT patients, data show steady weight gain (in pounds) throughout treatment between intervals, with the greatest percentage increase during month six of treatment (see Table 3). Overall, patients had a 35% increase in weight from baseline to discharge. The weight gained was significantly higher at discharge (M = 33.277, SD = 7.890) compared to baseline (M = 24.297, SD = 6.850; *t* (48) = −9.57, *p* < 0.001). Cohen’s *d* (1.367) indicated that this was a large-sized effect. A paired *t*-test was also conducted comparing baseline to 1 month and to 12 months; the results showed statistical significance and a large-sized effect.

For LD patients, the mean number of foods consumed at baseline was 2 (see Table 4). The variety of foods increased during subsequent treatment visits. At discharge, LD patients consumed a mean of 32 different foods. The number of foods consumed was significantly higher at discharge (M = 32.14, SD = 16.788) compared to baseline (M = 1.77, SD = 2.409; *t* (21) = ×8.188, *p* < 0.001). Cohen’s *d* (1.746) indicated that this was a large-sized effect. A paired *t*-test was conducted comparing baseline to after initial treatment as well; the results showed statistical significance and a large-sized effect.

Liquid-dependent patients consumed a mean of 15% of their needs via solids at baseline and 63% at discharge. 

For FS patients, the mean number of foods consumed was four at baseline (see Table 5). The number of foods increased in subsequent months and, at discharge, the mean number of foods consumed was 35. The number of foods consumed was significantly higher at discharge (M = 34.93, SD = 4.262) compared to baseline (M = 4.02, SD = 4.262; *t* (43) = −10.796, *p* < 0.001). Cohen’s *d* (1.627) indicated that this was a large-sized effect. A paired *t*-test was also conducted comparing baseline to after initial treatment; the results showed statistical significance and a large-sized effect.

From 77% to 100% of patients received prior therapy across medical diagnoses (see Table 1), and length of time in prior therapy ranged from 15 to 20 months. The treatment in our program ranged from 16–20 months. It took 31–37 visits in our program before discharge. In addition, 81% of all patients ate “typically” at discharge, with some variation across medical diagnoses (see Table 1). Eating “typically” was also high across feeding conditions (see Table 2, Table 3, Table 4 and Table 5), but the lowest percentage of patients eating “typically” was found in the TFD category (72.2%). The satisfaction surveys also yielded high treatment satisfaction by caregivers (*n* = 187; M = 4.8). For eating typically, patients in TFD category had more failures due to the fact that we were unable to completely transition to age-typical texture of foods by discharge. Children had the basic oral motor skills required to eat regular textured foods upon discharge; however, some were not eating the entire volume at that texture. Some still required pureed foods in combination of regular textured foods at discharge and, in those cases, the goal of typical eating would not be met. With some patients, the volume of regular-textured foods takes time to build, but further treatment may not be necessary. Furthermore, some failures can be attributed to poor treatment integrity. In some cases, families may not always be able to follow the treatment plan as prescribed, which may contribute to the lack of progress. 

## 4. Discussion

These data suggest that this Intensive Home-Based Interdisciplinary Pediatric Feeding Disorders Program was effective in decreasing tube-feeding dependence, increasing weight, decreasing liquid dependence, and increasing the variety of foods consumed. In addition, despite previous therapy, most patients were successfully discharged from our program. These preliminary data suggest that patients with a variety of medical diagnoses and feeding problems benefited from this model. However, the generalizability is limited to children who lived at home with at least one parent. Future studies should evaluate this model with children in foster care.

In addition, these data suggest that a home-based model can accomplish similar goals as day-treatment and inpatient feeding programs. This model considers environmental and interactional variables, such as family dynamics. Other advantages of this home-based program include that families prepare foods they want their child to eat based on cultural preferences; the child may feel more comfortable learning to eat in their home environment, and this eliminates the family’s need to travel. Although the home-based model presents many benefits, children who require more medical supervision should be treated in a clinic- or hospital-based program. Future studies should evaluate what subset of the population would benefit most from this treatment model. Since this mode of treatment relies on caregivers as change agents, future studies should evaluate whether caregiver attributes affect outcomes. Furthermore, it may be important to experimentally demonstrate whether high treatment integrity is related to positive outcomes. 

These data differ from previous outcome measure studies because the patients in this study were in treatment for an average of 17 months, with a decreasing number of visits, while previous studies evaluated short-term care to resolve the immediate feeding problem. To our knowledge, no other outcome measure studies have evaluated “typical” eating patterns at the end of treatment. Moreover, when working with children with feeding problems, it is critical to evaluate variables beyond eliminating tube feedings. If oral motor deficits exist, those, too, must be addressed in treatment, so the child is eating an age-typical diet at discharge. Generalization to other settings is also imperative. The patients in this study who were typical eaters ate in at least two locations at discharge. The goal was not only to meet caloric and nutritional needs, but to eat like age-typical peers, without a structured protocol. 

Although these data show promising results, there was no control group; therefore, a comparison to children with feeding disorders who did not receive treatment cannot be made. Future studies should compare data at each interval from a treatment group to a group of patients who are waiting to receive treatment. A between-subject comparison would further validate the home-based treatment model. The patients in this study had lengthy feeding therapy prior to their admission to this program without treatment success. Therefore, it is unlikely that treatment gains were solely due to maturation. Despite collecting data on prior therapies for these patients, we cannot say with certainty that our treatment was fully responsible for treatment success. 

Future studies may want to analyze growth parameters (weight/age, height/age and Body Mass Index(BMI)/age z-scores) for FTT patients prior to treatment, at baseline, during treatment and at discharge. Changes in BMI/age z-score could be used as an outcome measure to better reflect changes in nutritional status, in addition to percent weight change. In this study, height was not consistently collected due to a lack of equipment in the patient’s home. Future studies should evaluate both height and weight at regular intervals using the same equipment supplied by the treatment program. 

Furthermore, future studies should conduct nutritional analyses at baseline and discharge, instead of solely looking at the number of foods consumed. A nutritional analysis could provide more information with regards to vitamin or mineral deficits. Children with autism who display extreme food selectivity may be at risk for micronutrient deficiencies and a host of other medical issues resulting from their limited diet [33]; therefore, examining those risks before and after treatment would further validate this treatment model for food-selective children. 

## 5. Conclusions

In summary, the intensive home-based interdisciplinary model is an effective mode of treatment for children with pediatric feeding disorders. This may also be a more cost-effective approach in the long term. Prior to admission, these patients received treatments that may cost less but were not effective (i.e., still dependent on enteral feedings, poor weight gain, limited variety). There would be cost savings if patients were allowed to access these services from the start. The average estimated cost for a patient from evaluation to discharge in our program is USD 61,600.00, which is less than keeping a patient dependent on enteral feedings. This treatment model considers all aspects of eating, including the environment, parent–child interaction, and long-term success. This model provides a cost-effective approach to treating children with feeding problems, as opposed to gastrostomy tube feedings, years of ineffective interventions, or both. 

## Figures and Tables

**Figure 1 nutrients-14-04602-f001:**
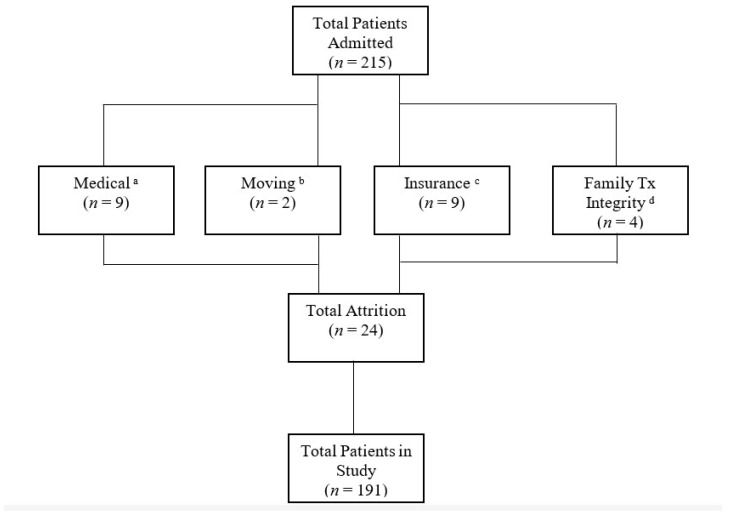
^a^ Patients failed swallow study or other medical concerns arose. ^b^ Family moved outside of geographical area. ^c^ Continuity of care to new insurance carrier denied. ^d^ Family unable to follow the treatment plan in the absence of the case manager. Tx= treatment.

**Figure 2 nutrients-14-04602-f002:**
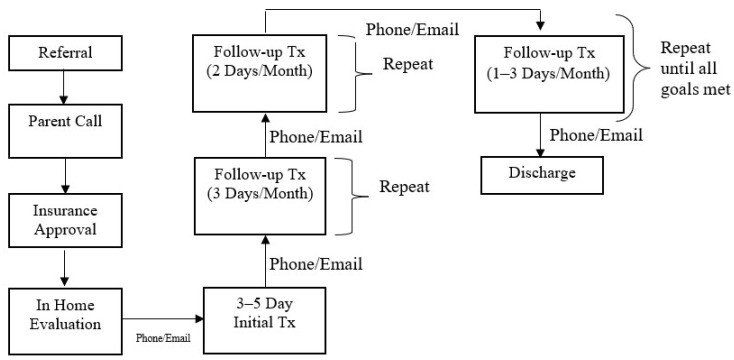
Referral and Treatment Process.

**Table 1 nutrients-14-04602-t001:** Categorization by Medical Diagnosis (*n* = 191) ^a^.

Medical Diagnosis	*n* ^b^	% Patients ^c^	% Typical Eating ^d^	% Tube Elimination ^e^	% Previous Therapy ^f^
Prematurity	53	27.7	79.6	75.0	90.9
Oral Motor Dysfunction	29	15.2	78.3	86.7	91.7
GER	127	66.5	76.4	77.6	90.0
FTT	91	47.6	77.5	69.8	84.2
Autism	35	18.3	88.2	88.9	77.1
DD	82	42.9	71.8	72.5	87.2
Other—Medical	131	68.6	74.4	78.5	90.4
None	6	3.1	100	N/A	100.0

Abbreviations: GER = gastroesophageal reflux; FTT= failure to thrive; DD = developmental disability; *n*= number of patients in the sample; N/A= not any. ^a^ Patients may fall into multiple medical diagnoses; however, they were only counted once in the total *n* = 191. ^b^ Number of patients who had each diagnosis. Patients with multiple diagnoses were counted more than once on this table. ^c^ Percent of patients who exhibited each diagnosis. Patients could have multiple diagnoses. ^d^ Percent of patients with each diagnosis who achieved age-typical eating at discharge. ^e^ Percent of patients with each diagnosis who had been tube-feeding-dependent, who achieved tube feeding elimination by discharge. ^f^ Percent of patients with each diagnosis who had received previous feeding therapy prior to our program.

## Data Availability

The data presented in this study are available on request from the corresponding author. The data are not publicly available due to protecting anonymity.

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
