# Peer review of "Evaluating Outcome Measure Data for an Intensive Interdisciplinary Home-Based Pediatric Feeding Disorders Program"

_nutrients, 2022, doi:10.3390/nu14214602_

Round 1

Reviewer 1 Report

"Evaluating Outcome Measure Data for an Intensive Interdisicplinary Home-Based Pediatric Feeding Disorders Program" is a relatively novel evaluation of intensive treatment for pediatric feeding disorders. Research to date predominately focuses on intervention delivered in highly structured intensive day-treatment or hospital based programs; however, there are additional research on home-based treatment programs. I recommend that authors review and reference the following articles for treatment of  children with food refusal and feeding tube or liquid dependence (Davis, A. M., Dean, K., Mousa, H., Edwards, S., Cocjin, J., Almadhoun, O., Jianghua, H., Bruce, A., & Hyman, P. E. (2016); Marinschek, S., Dunitz, S. M., Pahsini, K., Geher, B., & Scheer, P. (2014)), and food selectivity and/or ARFID (Burrell, Scahill, Nuhu, Gillespie, & Sharp, 2021; Johnson, Brown, Hyman, Brooks, Aponnte, Levato, & Smith, 2019; Sharp, W.G., Burrell, T.L., Berry, R.C., Stubbs, K.H., McCracken, C.E., Gillespie, S.E., & Scahill, 2019; Taylor, Blampied, & Roglic, 2020).

Authors of the this manuscript do a nice job describing the dependent measures and outcomes as it relates to nutrition; however, the greatest limitation to the manuscript is the lack of clarity on the actual treatment and parent traning methods. Authors state on page 4, "The feeding team provided in-person training to the case manager with regards to dietary modidfcication, oral motor exercises, behavioral treatmetns, and safety measures." Authors go on to say that a "structured protocol" was developed and once an "effective treatment was developed, caregivers were trained"; however, there is no information that details what the structured protocol entailed or how caregivers were trained.  If authors are only trying to convey that outcome measures can be collected in this sample, then the manuscript should be revised to focus on the importance of data collection in a home-based treatment. However, if authors want to demonstrate that their home based treatment program works to treat children with pediatric feeding disorders, then they must explain treatment in greater detail and strategies for training parents. Because this is a retrospective evaluation of treatment, it is unlikely that authors structured treatment delivery to be the same for each child presenting with each category of PFD; however, a treatment decision heirarchy and description of interventions should be described at a miniumum. More detail on the number of meals completed by therapist and by the parent and how parents were coached should also be described.

Author Response

  1. Add additional references.- These references have been added to the introduction. See lines 36, 43, 83-87
  2. Manuscript is lacking detail of the actual treatment and parent training.- More detail has been added to both treatment and parent training in the Methods. See pgs 4-5.

Reviewer 2 Report

Interesting paper reviewing the outcome of a pediatrics feeding disorder plan  based on ambulatory interventions instead of inpatient programs.

Some questions prior to accept the paper for publication:

1.       There is a mistake in the figure in line 64, regarding total costs of this kind of programs.

2.       Refer table 1 in the text. The same for figure 2.

3.       Please provide more information on the treatment satisfaction outcomes: were both parents consulted? How many answers from patients? Did you receive one o more questionnaires per patient?

4.       Report more information on the Method section regarding statistical analysis tools used.

5.       Results

a.       Provide mor demographic data on the sample. For instance, age.

b.       Data on weight gaining should be completed with changes in SD (not only reporting SD of the mean), as weight is a variable related to age.  This for groups TFD, FTT.

c.       Please, provide more information on failures if happened

Author Response

  1. There is a mistake in the figure in line 64, regarding total costs of this kind of programs.- This was revised on line 65.
  2. Refer table 1 in the text. The same for figure 2.- This was added see lines 105, 117, and 136.
  3. Please provide more information on the treatment satisfaction outcomes: were both parents consulted? How many answers from patients? Did you receive one o more questionnaires per patient?- This has been added to lines 265-268.
  4. Report more information on the Method section regarding statistical analysis tools used.- This has been added to lines 258-262.
  5. Provide mor demographic data on the sample. For instance, age.- This has been added to lines 106-108.
  6. Data on weight gaining should be completed with changes in SD (not only reporting SD of the mean), as weight is a variable related to age.  This for groups TFD, FTT. -I sent an email about the TFD patients since calorie data were presented and reviewer indicated then this question did not apply for TFD. We did add SD data for FTT category. These data were added to Table 3. 
  7. Please, provide more information on failures if happened.- This has been added to lines 357-366 and 383-387.

Round 2

Reviewer 2 Report

The authors have answered most of my suggestions. Although they have added weight in pounds and SD, it would have been ideal to have normalizes values (Z score).